# The Application of Phenolic Acids in The Obtainment of Packaging Materials Based on Polymers—A Review

**DOI:** 10.3390/foods12061343

**Published:** 2023-03-22

**Authors:** Beata Kaczmarek-Szczepańska, Sylwia Grabska-Zielińska, Marta Michalska-Sionkowska

**Affiliations:** 1Department of Biomaterials and Cosmetics Chemistry, Faculty of Chemistry, Nicolaus Copernicus University in Toruń, 87-100 Toruń, Poland; 2Department of Environmental Microbiology and Biotechnology, Faculty of Biological and Veterinary Sciences, Nicolaus Copernicus University in Toruń, 87-100 Toruń, Poland

**Keywords:** packaging materials, phenolic acids, antimicrobial properties, antioxidant properties, polymers

## Abstract

This article provides a summarization of present knowledge on the fabrication and characterization of polymeric food packaging materials that can be an alternative to synthetic ones. The review aimed to explore different studies related to the use of phenolic acids as cross-linkers, as well as bioactive additives, to the polymer-based materials upon their application as packaging. This article further discusses additives such as benzoic acid derivatives (sinapic acid, gallic acid, and ellagic acid) and cinnamic acid derivatives (*p*-coumaric acid, caffeic acid, and ferulic acid). These phenolic acids are mainly used as antibacterial, antifungal, and antioxidant agents. However, their presence also improves the physicochemical properties of materials based on polymers. Future perspectives in polymer food packaging are discussed.

## 1. Introduction

Packaging is an important type of material that provides a physical barrier and extends the shelf life of fresh food. It is used for different types of food for storage or transport purposes. Moreover, the selection of a type of food, or the producer, highly depends on the packaging material composition. While food-packaging materials are very important in everyday life, after we consume their contents, they remain as waste. Knowledge of the degradation processes is now higher than before, and consumers’ awareness and expectations concerning the environmental sustainability aspects of food packaging. Biodegradable raw compounds provide an opportunity to create eco-friendly materials dedicated to contact with food.

Another important factor that influences the type of raw material used to fabricate packaging is its antimicrobial activity against bacterial and fungal growth [1,2]. Both can be dangerous for humans to ingest, and packaging materials should not create an environment that promotes the growth of microorganisms. For that purpose, raw compounds can be enriched in active substances with antimicrobial activity [1,2,3]. Even if the basic raw material does not have antimicrobial activity, additives can be used that provide the final product with antibacterial and antifungal activity. Food-packaging materials with the addition of active substances that improve their physicochemical properties and provide them with specific properties such as antimicrobial, antifungal, or antioxidant are called active packaging materials. Characteristics of active packaging materials are presented in Figure 1.

Currently, phenolic acids are being explored as potential additives for packaging materials, because they are naturally sourced and have valuable antimicrobial effectiveness [4,5]. The advantage of using phenolic acids for the fabrication of packaging materials is that they may also act as cross-linkers in polymers and thereby improve their properties [6,7].

The aim of this review was to summarize current studies that include the testing of phenolic acids such as benzoic acid, sinapic acid, gallic acid, ellagic acid, cinnamic acid, *p*-coumaric acid, caffeic acid, and ferulic acid as active additives in packaging materials based on polymers.

## 2. Phenolic Acids

Phenols are a widely studied group of naturally derived compounds that are secondary plant metabolites. They may be found in plant-based foods, leaves, fruits, vegetables, etc. Phenolic acids contain a phenolic ring and an organic carboxylic acid function. They can be categorized by their origin, structural diversity, and biological function as bioactive compounds; these functional categories include flavonoids, tannins, phenolic acids, lignans, lignins, stilbenes, and coumarins. Over 10,000 phenolic compounds have been identified so far [8].

Phenolic acids are represented by cinnamic and benzoic acid derivatives. Cinnamic derivatives include *p*-coumaric acid, caffeic acid, and ferulic acid [9], and benzoic acid derivatives include gallic acid, ellagic acid, and sinapic acid [10]. Phenolic acids have been studied as cross-linkers of natural polymers. They improve the properties of obtained materials, mainly their mechanical parameters and stability.

Phenolic acids have been recognized as antioxidant and antimicrobial agents. Their antioxidant properties are attributed to their structure with hydroxyl groups that are capable of scavenging free radicals and conjugating double bonds, acting as a synergistic functional group [11,12].

## 3. Benzoic Acid Derivatives

Benzoic acid (BA) (Figure 2A) is also known as benzenecarboxylic acid, caboxybenzene, phenylformic acid, benzene formic acid, and phenylcarboxylic acid [13]. It is the simplest aromatic carboxylic acid. In this compound, the carboxylic group is bonded directly to the benzene ring. Benzoic acid is known as an antibacterial and antifungal preservative. It is commonly used in foods, food packaging, cosmetics, and medicines [13]. Benzoic acid can be obtained in fermented products through microbial metabolism, and it occurs naturally in plant and animal tissues [13].

T. Chowdhury and M. Das studied corn starch films with the addition of three different antimicrobials, including benzoic acid [14]. They obtained changed properties in the films after the addition of modifiers. BA acid reduced the tensile strength by 24% and adversely affected the optical properties of edible films. The authors observed a decrease in the whiteness index, an increase in the yellowness index, and a reduction in the surface gloss of starch films with 2.66% *w*/*w* addition of BA. They did not observe any changes in water vapor permeability after the addition of phenolic acid to matrices [14].

I.M. Lipatova et al. obtained and observed starch films with benzoic acid addition using one more component—chitosan—as a solubilizing additive [15]. The films based on starch, chitosan, and BA had greater tensile strength and a lower release rate of the preservative into model media than reference starch and sodium benzoate films. In addition, the antifungal activity of phenolic-acid-modified films was confirmed against *Aspergillus niger* [15].

Basic interactions between chitosan and benzoic acid were studied in 2011 by E.N. Fedoseeva and V.B. Fedoseev [16]. They observed physico-chemical properties of films depending on preparation conditions. Two types of materials were obtained: biphasic films containing fractal inclusions in a transparent matrix and homogeneous optically transparent films, where the presence of ionic interaction between BA and chitosan was confirmed by infrared spectroscopy [16].

I. Brink et al. composed chitosan and whey protein films with the addition of benzoic acid as an antimicrobial substance, with incorporated cranberry and quince juice [17]. They observed the antimicrobial activity of edible films applied to fresh-cut turkey pieces. Based on the results, these films stopped the development of pathogenic microorganisms *Salmonella typhimurium*, *Escherichia coli*, and *Campylobacter jejuni* and stopped the microbiological deterioration of the turkey meat for at least six days [17].

Films based on protein from Argentine anchovy (*Engraulis Anchoita*) were incorporated with benzoic acid [18]. The films showed antibacterial activity against *Escherichia coli*, *Salmonella enteritidis*, and *Listeria monocytogenes*, but they did not show antibacterial activity against *Staphylococcus aureus.* Additionally, for materials with 1.5% *w*/*w* BA, decreased tensile strength and increased elongation at break were noticed [18].

In addition to food-packaging biopolymer-based materials with the addition of BA, materials based on other polymers were also obtained. BA was used as the additive to materials based on poly(lactic acid) [19], polyethylene [20], and poly(ethylene-co-methacrylic acid) [21]. In poly(lactic acid) films, BA was used as an ingredient incorporated into β-cyclodextrin-grafted TiO_2_ nanoparticles, which were dispersed in biodegradable matrices [19]. In one study on low-density polyethylene (LDPE), propionic acid, benzoic acid, and sorbic acid were added as antimycotic agents. None of the listed additives inhibited mold growth when the films came into contact with inoculated media [20]. However, when benzoic anhydride was incorporated into matrices, films exhibited antimycotic activity during contact with media and cheese. Poly(ethylene-co-methacrylic acid) films with addition of BA showed dominantly antimicrobial properties in fungal growth inhibition tests (*Aspergillus niger* and *Penicillium* sp.) [21].

The derivatives of BA (Figure 2A) include sinapic acid (Figure 2B), gallic acid (Figure 2C), and ellagic acid (Figure 2D), which will be described in the next part of the review.

### 3.1. Sinapic Acid

Sinapic acid ((*2E*)-3-(4-Hydroxy-3,5-dimethoxyphenyl)prop-2-enoic acid, SA) also known as sinapinic acid, is a small, naturally occurring hydroxycinnamic acid. It possesses 4-hydroxyl and 3,5-dimethoxy substitution in the phenyl group of cinnamic acid [22,23]. Sinapic acid is widely used due to its antioxidant, anti-inflammatory, antibacterial properties and anxiolytic-like effects. It occurs in rye, fruits, and vegetables, and it is one of the main phenolic compounds of canola and rapeseed [22,23]. While sinapic acid is a common acid in the plant kingdom and is widely used in various applications, reports about its use as a modifier for biopolymers are uncommon.

The small number of available studies on the influence of SA on the physico-chemical properties of polymeric materials report a number of characteristics in this phenolic acid that prompted us to include a section on it in this review.

K. Crouvisier-Urion et al. studied films based on lignin and chitosan with three phenolic acids: ferulic acid, coumaric acid, and sinapic acid. They observed antioxidant activity, and SA acid was the best antioxidant [24].

D. Rabiej-Kozioł et al. studied changes in the quality of rapeseed oil with sinapic acid ester–gelatin/poly(vinyl alcohol) films during storage [23]. They prepared gelatin/poly(vinyl alcohol) active films with the addition of three sinapic acid esters: ethyl sinapate, octyl sinapate, and cetyl sinapate. The authors studied the antioxidant activity and the thickness of films. Based on their results, the addition of each sinapic acid ester resulted in a significant enhancement of the antioxidant activity of the gelatin/poly(vinyl alcohol) films [23]. Additionally, it was clearly visible that the addition of sinapic acid esters resulted in a significantly increased thickness of the films in comparison with the gelatin/poly(vinyl alcohol) non-modified samples. The authors also observed that the carbon chain length of esters added to the film affected their thickness: a longer chain of ester resulted in an increased thickness of films [23].

SA was detected in the leaf extract of *Pistacia Terebinthus*, which was blended with chitosan, and edible chitosan films were obtained [25]. The films had high antioxidant activity and antimicrobial activity against nine food-borne pathogens (*Escherichia coli*, *Staphylococcus aureus*, *Proteus microbilis*, *Proteus vulgaris*, *Pseudomonas aeruginosa*, *Enterobacter aerogenes*, *Bacillus thuringiensis*, *Salmonella enterica*, *Streptococcus mutans*) [25].

### 3.2. Gallic Acid

Gallic acid (3,4,5-trihydroxybenzoic acid, GA) isa natural phenolic acid that can be isolated from black tea, mango, banana, berries, clove, thyme, or chestnut [26]. It is a secondary metabolite present in most plants, and it exhibits antioxidant, antimicrobial, antifungal, anti-inflammatory, anticancer, and antiulcerogenic properties. It is commonly used in medicinal areas, the skin and leather industries, photographic usages, and food-packaging applications [27].

Gallic acid has been used as a cross-linker to obtain chitosan-based materials [28,29,30,31,32,33]. Chitosan/GA coating showed antioxidant and antimicrobial activity. Such materials may be used to develop a practical hurdle technology in the preservation of fresh pork, resulting in the improvement of food safety and quality [28]. Antimicrobial activity against *Escherichia coli*, *Salmonella typhimurium*, *Listeria innocua*, and *Bacillus subtilis* was proven for chitosan/GA films. The use of GA to prepare chitosan-based films also resulted in the improvement of barrier properties and reduction of water vapor and oxygen permeability [29]. Additionally, high antioxidant activity with DPPH was observed for chitosan/GA films after 4 days [30].

Chitosan-based films containing ZnO particles loaded with GA were prepared by S. Yadav et al. [34]. The presence of ZnO with GA improved the antioxidant and antimicrobial activity. Moreover, the improvement of mechanical properties and water vapor permeability rate was observed. Due to their improvement of the shelf life of food products, such coatings can be proposed for food packaging [34].

Starch-based materials were cross-linked with GA by non-covalent interactions [35]. The addition of GA resulted in the reduction of accessibility of starch molecules to digestive enzymes. Thus, GA–starch complexation could be beneficial in controlling the digestion behaviors of starch-based products [36]. Bioactive films were fabricated from starch (potato by-product) with GA addition. An improvement of antioxidant and antimicrobial activity was observed. Moreover, GA acted as a plasticizer in high concentrations of gallic acid (0.2 g gallic acid/g cull starch) [37].

It was reported by J. Promsorn and N. Harnkarnsujarit [38] that an increasing amount of GA added to thermoplastic starch plasticized and facilitated starch melting. Thus, it may be assumed that GA added to the starch not only acts as an antioxidant and antimicrobial agent but also improves the applicable properties of materials. Furthermore, the minimum residual oxygen and rates of scavenging were correlated linearly with the content of GA [38].

Gallic acid was also used as a modifier for chitosan and starch films [39,40]. They have been obtained by subcritical water technology [39]. The formation of ester bonds, hydrogen bonds, and electrostatic interactions between chitosan, starch, and GA has been observed [39,40]. These kinds of materials were characterized by reduced elongation, reduced film solubility in water, and reduced water vapor permeability, while the improved tensile strength and excellent transparency of films were detected [39].

In addition to the biopolymer–gallic-acid connections mentioned above, gelatin [41], chitosan/gelatin [42], gelatin/casein [43], and cellulose/kappa-carrageenan [44] films modified by gallic acid have been obtained, and their physico-chemical properties have been studied. In gelatin/gallic acid materials, the aldehyde group of GA reacts with the amino group in the gelatin molecule to form a stable covalent bond [41]. An increased degree of cross-linking in the gelatin films was observed. The gelatin/gallic acid matrices were characterized by stronger tensile strength, higher elongation at brake, good antioxidant activity, biodegradability, and antibacterial properties against *Staphylococcus aureus* and *Escherichia coli* [41]. L. Rui et al. observed that with the increasing concentration of GA, in chitosan/gelatin/gallic acid films, there was a reduction in transmittance, elongation at the brake, and water vapor permeability [42]. In gelatin/casein/gallic acid films, the films had a semi-crystalline structure, with enhanced thermal stability, more homogenous surfaces, increased Young modulus and tensile strength, and reduced elongation at the brake than pristine films [43]. Additionally, kappa carrageenan and cellulose nanofibers were grafted with GA. Biopolymer films with antioxidant activity were obtained and studied [44].

A gelatin-based coating containing GA and tannic acid was incorporated into the poly(lactic acid). Phenolic acids interacted with gelatin and the mixture was placed as plasticized coating (PLA). The surface hydrophilicity slightly decreased as a result of the cross-linking by phenolic compounds. Moreover, the mechanical properties of PLA were maintained after coating, and their barrier properties were highly improved [45].

Gallic acid was used as a natural additive in high-density polyethylene (bio-HDPE) formulation. HDPE was firstly melt-compounded and then loaded by GA. Obtained films were studied as potential food packaging. Delayed onset oxidation temperature (OOT) was observed for films containing GA, as well as an increase in UV light stability, which may cause lipid oxidation in food products [46].

Gallic acid was grafted to poly(ɛ-caprolactone) (PCL) by UV photo-induction method [47]. A significant change of mechanical properties was observed after the GA grafting. However, the GA grafting resulted in the increase of roughness of the surface of the fabricated films. PCL films were determined as hydrophobic by contact angle measurement. After grafting, an increase in wettability was observed [47].

### 3.3. Ellagic Acid

Ellagic acid (4,4′,5,5′,6,6′-hexahydroxydiphenic acid 2,6,2′,6′-dilactone, EA) is a polyphenol with anticancer and excellent antioxidant effects and can be found in numerous fruits. It is derived from gallic acid metabolism [48,49,50]. Ellagic acid occurs in pomegranates, strawberries, blackberries, raspberries, blueberries, nuts, seeds, and green tea. It is commonly known that ellagic acid is characterized by interesting biological activities, including antioxidant activity, antibacterial activity, and UV-barrier properties [50,51].

There is a limited number of scientific reports concerning ellagic acid as a modifier to biopolymeric films with potential application in the food packaging industry. However, the advantages of this phenolic acid and the information found in the reports cited below prompted us to include a section about it in this review.

The films obtained from chitosan and EA were homogeneous, translucent, and flexible [50]. The authors applied 0.5, 1, 2.5, and 5% *w*/*w* addition of EA, and chitosan/ellagic acid films were obtained by solvent-casting method. The obtained materials showed high mechanical parameters and high thermal stability. UVA- and UVB-barrier properties, moderate water vapor permeability, and high antioxidant activity were observed in chitosan/ellagic acid films [50]. Antibacterial properties were also evaluated, and the results showed antibacterial activity against to gram-positive (*Staphylococcus aureus*) and gram-negative (*Pseudomonas aeruginosa*) strains. Based on these results, the authors stated that chitosan/ellagic acid films could be potentially used for active eco-friendly packaging [50].

In 2018, J.M. Tirado-Gallegos et al. [52] prepared and characterized apple starch films incorporated with EA. Three different concentrations of EA were added (0.02%, 0.05%, and 0.1%), and films were obtained by casting method. The mechanical parameters, water vapor permeability, thermal stability, and optical and morphological properties of the obtained films were evaluated. It was found that films with 0.05% and 0.1% EA were characterized by rough surfaces, and insoluble EA particles were observed in these films [52]. The addition of phenolic acid to matrices modified the values of mechanical parameters: tensile strength, elastic modulus, and elongation at break. Additionally, films were capable of blocking UV light and showed higher antioxidant activity than control films (without EA) [52]. However, despite these positive properties, it would be worthwhile to investigate new strategies for the solubility of EA in biopolymeric solutions and the antibacterial activity of potential active packaging materials.

Ellagic acid was detected as one of the main ingredients of guava leaf extract, which was blended with sodium alginate to obtain green packaging films [53]. A compact structure in the composite films was observed, which was formed from the intermolecular hydrogen bonding between the guava leaf extract and sodium alginate. In addition, enhanced antioxidant and antibacterial activity, tensile strength, water solubility, and water barrier properties could be observed after adding the guava leaf extract to sodium alginate matrices. Furthermore, a decrease in the moisture content and elongation at the break of sodium alginate/guava leaf extract has been observed [53].

Using EA as a dietary supplement in powder, liquid forms, or capsule is much more common than using it as a food-packaging film modifier [52]. EA is also used in the biomedical field [54,55,56].

## 4. Cinnamic Acid Derivatives

Cinnamic acid (3-phenylprop-2-enoic acid, 3-phenylacrylic acid, CA) (Figure 3A) is a natural aromatic carboxylic acid [57]. The main sources of cinnamic acid are plants, such as *Panax ginseng* and *Cinnamomum cassia* (Chinese cinnamon), whole grains, vegetables, fruits, and honey [57,58]. It can occur in cis or trans configuration, due to the presence of acrylic acid group substituted on the phenyl ring, but the trans configuration is the most common. It is generally known that CA exhibits a lot of excellent properties, e.g., antioxidant, antimicrobial, anti-inflammatory, antidiabetic, anticancer, neuroprotective [57]. It can be used as an additive to food-packaging materials and fragrant ingredients in cosmetics, toiletries, and detergents [57,59,60,61,62].

R. Ordoñez et al. studied cassava starch (CS) [63] and poly(lactic acid) (PLA) [64] films with the addition of CA. They used thermal processing to obtain thin materials in both cases. Starch films with the addition of CA were less water-soluble, more extensible, and less resistant to break than non-modified materials. The addition of CA to CS films had no significant influence on the barrier properties of the films [63]. Additionally, the starch films showed antibacterial activity against *Escherichia coli* and *Listeria innocua*, where they were more active against *Listeria innocua*. The antibacterial assays were studied in culture medium and food systems: fresh-cut melon and chicken breast [63]. In PLA films, the authors reported that the addition of CA to films slightly worsened the mechanical properties, and the films were less stiff, less resistant to break, and less extensible than native films. The addition of 1% and 2% *w*/*w* of CA was studied, and the 2% addition of CA improved the water vapor barrier capacity of the films. The thermal stability of PLA films was higher in CA-modified films. Furthermore, the release kinetics of CA from the polymeric matrix were observed, and no quantitative release was detected into aqueous and non-polar media, while limited release was observed into ethanol 50% *v*/*v*. Based on these results, no antibacterial activity against *Escherichia coli* and *Listeria innocua* was observed [64].

The same authors [65] studied PLA films with a higher CA addition and two different processing methods. The first method was thermal processing, where 5 and 10% *w*/*w* were successfully incorporated into PLA film. The second method was casting (ethyl acetate used as a solvent to PLA), with which 3% *w*/*w* of CA was incorporated into PLA films without the crystallization effect of cinnamic acid [65]. The authors observed the microscopic structure of obtained films and evaluated their thermal, mechanical, barrier, optical, and antibacterial properties. They also observed that the incorporation of CA reduced the PLA film’s stiffness and resistance to breaking in both processing methods [65]. Additionally, the water vapor and oxygen barrier capacity of the films was better for PLA films with CA addition than for PLA pristine films. In this research, there was also no significant antibacterial activity observed against *Listeria innocua* in the in vitro assays because only the CA molecules near the surface of the films were released into the medium [65]. Another article, published by the same group of scientists, concerned the antibacterial properties of cinnamic acid, ferulic acid, and ester of ferulic acid incorporated into starch and PLA monolayer and multilayer films [66]. This group observed that starch monolayers with cinnamic and ferulic acid showed notable growth inhibition capacity against *Escherichia coli* and *Listeria innocua*, while PLA monolayers containing the higher content of these acids of derivative of ferulic acid did not exhibit antibacterial capacity. They observed that, regardless of the film preparation method and the limiting factor for the compound, the release that was caused by reduced molecular mobility in the PLA matrix could be the reason for the subsequent weak antimicrobial effect [66].

Materials based on PLA and starch mixture with the addition of cinnamic and ferulic acid were also obtained and studied [67]. The procedure of film preparation was complicated, and it included the lamination of PLA and starch as three-layer assemblies (with two PLA outer layers, which were responsible for protection the interior of the starch from moisture) and incorporation of cinnamic or ferulic acid onto the PLA surface (electrospinning or the pulverization of the cinnamic and ferulic acid solutions was used) [67]. The antibacterial tests of the obtained active films showed effective growth inhibition of *Escherichia coli* and *Listeria innocua*. In this study, CA showed greater antibacterial activity against *Listeria innocua* than ferulic acid [67].

N. Benbettaïeb et al. [68] prepared gelatin/chitosan (1:1 *w*/*w*) bioactive packaging films with CA and GA, and they studied the release kinetics of them in food simulants having different water activities and viscosities [68]. They found that the nature and content of the food simulant affects the release kinetics parameters, structure of the films, and bioactivity of them. The films with CA addition showed antioxidant activity, but it was lower than the antioxidant activity of gelatin/chitosan films with GA addition [68].

Poly(vinyl alcohol) (PVA) can also be adapted to produce films with potential use in food-packaging applications [69,70]. Johana Andrade et al. [69] prepared PVA films with the addition of cinnamic acid and ferulic acid. Firstly, two kinds of PVA of different molecular weights and degrees of hydrolysis were used in the preparation of the films [69]. Secondly, in another article, the same authors prepared films by two processing methods: casting of the polymeric aqueous solutions and melt blending with compression molding [70]. Homogenous film microstructure was observed for partially acetylated PVA with the addition of CA [70]. They also observed that fully hydrolyzed, high-molecular-weight PVA films and added CA had better mechanical and barrier properties than films prepared based on partially hydrolyzed PVA [69]. The addition of CA to the films had a positive antioxidant response [70]. Moreover, films with added CA were characterized by better barrier properties against native PVA films [70]. In addition to this, the presence of CA in the PVA films clearly induced the inhibition of *Listeria innocua* growth [70].

Eco-friendly food packaging based on cellulose with cinnamic acid has been prepared by X. Rumeng et. al. [71]. They used reduction of the degree of polymerization of cellulose and the degree of substitution of cinnamate in the processing method of materials. The authors obtained transparent films characterized by biodegradability, hydrophobicity, biosafety, and thermoplasticity. The materials were safe for human epidermal cells, and their transition temperature and thermal flow temperature could be adjusted [71].

In addition to the cases described above, CA has also been used to modify konjac glucomannan/PLA micro-films, which exhibited great antibacterial activities against *Staphylococcus aureus* and *Escherichia coli*, excellent mechanical properties, thermal stability, hydrophobicity, and good swelling degree [72].

Some studies in the literature discuss the use of cinnamic acid derivatives (potassium cinnamate) for the modification of hemicellulose/poly(vinyl alcohol) blends. This kind of material showed increased elongation at break, moderate oxygen barrier properties, good thermal stability, good UV barrier properties, and antibacterial activity against *Escherichia coli* bacteria [73].

The derivatives of cinnamic acid (Figure 3A) include *p*-coumaric acid (Figure 3B), caffeic acid (Figure 3C), and ferulic acid (Figure 3D), which will be described in the next part of the review.

### 4.1. p-Coumaric Acid

*p*-Coumaric acid ((*2E*)-3-(4-hydroxyphenyl)prop-2-enoic acid, PA) is a unique chemical structure and is known as an antioxidant and antimicrobial agent with low toxicity [74]. It can be found in various edible plants, such as tomatoes, carrots, and cereals [75].

Hydrocolloid-based films were obtained from *p*-coumaric acid with chitosan and fish gelatin. Those films showed lower antioxidant activity than chitosan/fish gelatin/caffeic acid films [76]. They showed antioxidant and antimicrobial activity against *Escherichia coli*, *Salmonella*, *Bacillus subtilis*, and *Staphylococcus aureus*. The thermal stability was also improved by the addition of *p*-coumaric acid to chitosan thin films [77].

The *p*-coumaric acid-modified chitosan was incorporated into the poly(vinyl alcohol)/starch films. This resulted in the increase of tensile strength from 15.67 MPa to a maximum of 24.32 MPa. The film prevented water diffusion and exhibited excellent swelling, water vapor transmittance, and antioxidant activity. An improvement in thermal stability was also observed. Moreover, films showed antimicrobial activity against both gram-negative and gram-positive bacteria [78].

In the article by Young H. et al. [79], chitosan films were functionalized with three different hydroxycinnamic acids, including PA. They confirmed that hydroxycinnamic acids were conjugated with chitosan through amide and ester bonds [79]. Chitosan/*p*-coumaric acid films were characterized by a more compact microstructure, higher UV light barrier ability, thermal stability, antioxidant activity, water vapor barrier ability, and mechanical strength. Furthermore, antibacterial activity against *Escherichia coli*, *Salmonella typhimurium*, *Staphylococcus aureus*, and *Listeria monocytogenes* was detected [79].

PA acid was also used as a modifier to cellulose nanocrystals, which were uniformly dispersed in the pectin matrix to improve coating barrier properties. *p*-coumaric acid introduced antioxidant properties to the cellulose nanocrystals [80].

Chitosan/poly(vinyl alcohol) membranes enriched with PA were fabricated by solution casting method [81]. The membrane showed a hydrophilic character. The addition of phenolic compound resulted in the increase of volume and weight-swelling degree.

### 4.2. Caffeic Acid

Caffeic acid ((*2E*)-3-(3,4-dihydroxyphenyl)prop-2-enoic acid, CFA) may be isolated from coffee, mint, oregano, rosemary, thyme, coriander, cardamon, blueberry, yerba mate, mango, and banana [26]. It has two hydroxyl groups, para- and ortho-substituted, on an aromatic ring. Therefore, its activity is higher than gallic acid. Caffeic acid has been reported as a natural antioxidant used in food packaging [82].

Films based on chitosan and cellulose were fabricated with CFA addition to prepare active food-packaging materials by S.H. Yu et al. [83]. Chitosan/cellulose-based materials with caffeic acid showed higher antioxidant and antimicrobial activity than films without phenolic acid addition. CFA was continuously released from the materials and showed an inhibitory effect on the lipid oxidation of menhaden. Such materials can be used as food packaging due to their reduced lipid oxidation ability [83].

N. Benbettaied et al., obtained chitosan–fish-gelatin films containing different phenolic acids [76,84], including CFA as a natural antioxidant. Its presence improved the mechanical properties of films, as caffeic acid also acts as a cross-linker of chitosan as well as gelatin [84].

C. Nunes et al. [85] studied materials based on chitosan grafted with CFA cross-linked by genipin. Films showed antioxidant activity and lower solubility. The properties such as wettability, thermal stability, and mechanical properties did not change significantly. Chitosan–caffeic-acid film properties were compared with chitosan–gallic-acid ones. The results showed that those containing CFA exhibited ideal apparent properties with less yellowness and better mechanical parameters [31].

Materials from fish gelatin with the addition of ferulic acid and caffeic acid were also obtained and characterized [86]. The results showed that CFA is a more effective phenolic acid than ferulic acid. This is because, in a comparison of physico-chemical properties, CFA had the highest effect in decreasing solubility, water vapor permeability, and oxygen permeability [86].

The CFA was added to the poly(vinyl alcohol-co-ethylene) (EVOH) as an active additive, and materials in film form were fabricated by solvent-casting process [87]. Films were studied for potential application as packaging. The addition of caffeic acid resulted in the increase of mechanical parameters. Moreover, films with CFA showed positive radical scavenging activity. The amount of 5% of active additive was considered a strategic amount to guarantee the structural and functional characteristics.

Caffeic-acid-grafted chitosan/poly(lactic acid) films were obtained and characterized by Zhou et al. [88]. It was assumed that compared to traditional polyethylene packaging, fabricated films would demonstrate lower permeability, higher fluidity, and a stronger ability to maintain free water. Materials based on caffeic-acid-grafted chitosan/poly(lactic acid) delayed the decreased rate of unsaturation value and phospholipids of *Agaricus bisporus* during storage. They enhanced the postharvest quality of *Agaricus bisporus* by regulating membrane lipid metabolism [88].

Microfibrous polycaprolactone containing different amounts of CFA materials was obtained by electrospinning method [89]. The addition of caffeic acid did not affect the diameter of fibers or their structure. Fibers showed hydrophobic behavior. The most suitable concentration of CFA was determined as 0.10% due to the absence of anomalies on the surface of the polycaprolactone microfibers.

There is a lack of information about the degradation of polymer-based materials containing caffeic acid. In our opinion, research on this topic should be carried out to determine their bio-degradability and eco-friendliness.

### 4.3. Ferulic Acid

Ferulic acid (4-hydroxy-3-methoxycinnamic acid, FA) can be found in many plants. It is commonly found in commelinid plants, such as rice, wheat, oats, or pineapple [90]. Additionally, ferulic acid can be obtained from grasses, grains, vegetables, flowers, fruits, leaves, beans, coffee bean, artichoke, peanut, and nuts. Ferulic acid is insoluble in water at room temperature, but it is soluble in hot water, ethanol, ethyl ether, and ethyl acetate [90].

Ferulic acid is commonly used as an additive to films with potential food-packaging applications [90]. Apart from that, it is also used as an antidiabetic and antiaging agent, anticancer agent, food preservative, photoprotective constituent in cosmetics, indicator of environmental stress in plants, and active agent in biological contexts [71]. It has such wide application due to having the following properties: antioxidant, antiallergic, hepatoprotective, anticarcinogenic, anti-inflammatory, antimicrobial, antiviral, and antithrombotic [90].

B.B. Yerramathi et al. used ferulic acid as a crosslinking agent to modify sodium alginate films [91]. They reported that due to the FA crosslinking, the sodium alginate films were transparent, homogenous, thermally stable, and more rigid than non-modified films. The films exhibited antioxidant activity and did not show inhibition against *Klebsiella pneumonia* and *Salmonella enteric*, which are the most common food-spoiling bacteria [91].

Chitosan has been modified by phenolic acids, including FA [92,93]. Thin films based on chitosan with added FA showed antibacterial activity against gram-positive (*Staphylococcus aureus*) and gram-negative (*Escherichia coli*) strains [92]. Additionally, cross-linking of chitosan films by FA improved mechanical properties and thermal stability due to hydrogen bonds between chitosan and ferulic acid. The authors also observed that the sterilization of chitosan/ferulic acid films via exposure to UVC light was effective, and they proposed 1 h exposure as a standard sterilization process for food-packaging materials [92]. However, other authors who have studied the properties and preservative effects on *Penaeus vannamei* of chitosan films modified by phenolic acids, including FA, observed a negative effect on their mechanical properties and a positive effect on their bioactivity [93].

Ferulic acid was also added to a mixture of sodium alginate and chitosan [94]. Layer-by-layer self-assembly methods have been used to obtain sodium alginate/chitosan/ferulic acid films. These kinds of materials were characterized by high tensile strength, good light-blocking performance, hydrophobicity, thermal stability, and lower water vapor transmission rate and swelling degree [94]. The results obtained by K. Li et al. showed strong interactions between the amino, carboxyl, and hydroxyl groups of the ferulic acid, sodium alginate, and chitosan [94].

Pullulan-based composite films with bacterial cellulose and FA were studied by Z. Ding et al. [95]. The films were obtained by casting method, and a physico-chemical characterization of materials was prepared. Firstly, studies were done in a solution system. Bacterial cellulose and FA were uniformly dispersed in pullulan solution to form uniform and dense composite films [95]. The obtained materials were characterized by antioxidant activity, superior anti-fogging activity, and high mechanical strength and thermal stability. Additionally, pullulan/bacterial cellulose/ferulic acid films showed water, oxygen, and carbon-dioxide barrier performances [95].

Melt-processed bioactive ethylene vinyl alcohol copolymer (EVOH) films incorporated with FA (0,25; 0,5; 0,75; 1 wt.%) were prepared and characterized by Alejandro Aragón-Gutiérrez et al. [96]. Structural, morphological, thermal, and functional characterization of obtained films have been prepared. The authors reported that FA could be used as an additive to active food packaging due to its antimicrobial and antioxidant properties, together with enhanced ductility, thermal stability, and UV-blocking effect [96]. Additionally, a degradation effect was observed for the melt-compounding film method, which is an advantage in such applications. The materials exhibited antimicrobial activity against gram-positive (*Staphylococcus aureus*) and gram-negative (*Escherichia coli*) bacteria, and they showed high effectiveness in radical-scavenging inhibition (DPPH method was used) [96].

In a mixture of PLA and poly(butylene adipate-co-terephthalate) (PBAT) in a 98:2 ratio, addition of FA (1, 5, and 10 wt.%) was explored [96] and exhibited stronger antibacterial efficacy against *Listeria monocytogenes* and *Escherichia coli* [97]. Like the films discussed above [96], these also showed a slight tint of yellow and significant UV-light barrier property after the addition of FA to matrices [97]. The authors also noted enhanced tensile strength, fracture failure, elasticity, and thermal stability of the ferulic-acid-modified films [97].

In thermo-processed PLA films, the incorporation of FA did not notably affect functional properties but increased the thermal degradation temperature of PLA matrices [64]. Very limited release of FA from PLA films was detected in food simulants, and no significant antibacterial activity was observed when 1 or 2% *w*/*w* of FA was used [64]. However, for films obtained from cassava starch, ferulic acid inhibited *Listeria innocua* and *Escherichia coli* bacterial growth in culture medium tests and in chicken breast [63]. Furthermore, films with FA were more extensible than non-modified films and cinnamic-acid-modified starch films [63]. As described above (Section 4), starch monolayers with CA and FA showed growth inhibition capacity against *Escherichia coli* and *Listeria innocua*, while PLA monolayers with higher addition of these acids or methyl ester of ferulic acid did not exhibit antibacterial capacity [66].

In the next study, ethylene vinyl acetate copolymer (EVA) and low-density polyethylene (LDPE) were blended in various compositions, and FA in 2, 4, 6, 10, and 14% *w*/*w* was added [98]. The release rate of ferulic acid was studied. The results showed that controlled release of ferulic acid was possible based on changing the polymer blend composition [98].

Ferulic acid was used as the active substance in PLA:PVA [99] and PLA:starch [67] three-layer films by Johana Andrade et al. and Ramón Ordoñez et al., respectively. The addition of phenolic acid to matrices had a slight effect on the functional properties of the three-layers films. Nevertheless, for the PLA:PVA mixture, the authors observed effectiveness in the active substance, which controlled the microbial growth of beef meat over 17 days of cold storage. Their materials successfully promoted shelf-life extension and meat preservation [99]. In PLA/starch films, antibacterial tests showed effective growth inhibition of *Escherichia coli* and *Listeria innocua*. Additionally, as described above (Section 4), FA showed lower antibacterial activity against *Listeria innocua* than CA [67].

Apart from the above-mentioned single polymers and their blends, a combination of poly(lactic acid) and poly(3-hydroxybutyrate-co-3-hydroxyvalerate) (PHBV) with added FA was also tested [100]. This kind of ferulic-acid-modified film had higher glass transition temperature, increased thermal stability, stiffer and more resistant structure, and improved oxygen and water vapor barrier capacity. As for the antibacterial properties, in this case, gram-positive (*Listeria innocua*) and gram-negative (*Escherichia coli*) bacteria were used for testing, and the materials exhibited antibacterial activity [100].

Additionally, ferulic acid was a modifier of active films based on zein [101], soy protein [102], and myofibrillar proteins of bigeye snapper (*Priacanthus tayenus*) [103]. In zein films, FA was used as a plasticizer, and its addition to films eliminated their brittleness. However, the zein/ferulic acid films lost structural integrity when hydrated in distilled water and showed extreme swelling [101]. The addition of ferulic acid to soy protein films caused color and transparency changes and the reduction of water vapor permeability and water solubility [102]. Color and transparency changes and excellent barrier properties against UV light (at the wavelength of 200–800 nm) were observed for films based on myofibrillar proteins of bigeye snapper [103].

In addition to ordinary ferulic acid, its derivative was also used to modify biopolymers [104]. Laccase was added to FA solution to initiate the oxidation process, and laccase-oxidized ferulic acid was obtained. Next, collagen films were immersed in laccase-activated ferulic acid solution. The results obtained from the physico-chemical characterization of the films showed that they have potential applications as active food-packaging materials [104]. The films were characterized by good mechanical properties, thermal stability, resistance to enzyme degradation, antioxidant activity, and antibacterial activity against *Escherichia coli* and *Staphylococcus aureus* [104].

## 5. Summary of the Discussed Materials—Properties and Applications

In this section, the tables that follow demonstrate the active packaging materials that have been presented and their main properties. While most are biopolymer materials modified with phenolic acid addition, materials based on mixtures of natural and synthetic polymers and those based on other polymers are included as well. Table 1 shows the properties and applications of biopolymeric materials with phenolic acid modifications. Table 2 presents the properties and applications of the other discussed materials with phenolic acids modification.

## 6. Future Perspectives

Materials based on polymers with phenolic acid addition can be considered active, as their antioxidant and antimicrobial activity have been proven. Moreover, research has shown that the addition of phenolic acids improves the mechanical parameters of obtained materials as cross-linkers of polymers.

Materials based on polymers with phenolic acids have potential for use as food packaging. As the main advantage of such systems is that the components are naturally sourced, they are considered eco-friendly; however, there is a lack of studies related to their influence on the environment. Antimicrobial agents, such as phenolic acids, also show activity on microbial presence in the soil, air, and water. In our opinion, more studies of these materials’ degradation should be carried out on their behavior in natural conditions as well as their influence on the microbial flora. It is necessary to determine the by-products of the degradation process and if their presence in the natural environment is harmful.

The Influence of polymer/phenolic materials on the food that is stored in them also requires further study. The type of food for which the obtained materials are used must be identified. Therefore, there is a need to study the properties of food after their storage with the proposed packaging materials. In summary, materials based on polymers with phenolic acids addition are promising for food packaging, but further studies should be carried out to consider their effect on the natural environment and their application properties in food storage.

The large-scale industrialization of films based on polymers with phenolic acids is another matter of concern. It is important to find an effective method for scaling production from from the laboratory to industry. Without it, the application of the discussed films as food packaging will not be possible.

## 7. Conclusions

This review can be used by a wide group of scientists, researchers, and companies that are working to obtain appropriate, biodegradable food-packaging materials with antibacterial, antifungal, and antioxidant properties. The blending of polymers with phenolic acids, and obtaining biodegradable food packaging materials, especially films, can improve the physicochemical properties of matrices. Natural polymers, as well as synthetic ones, can be effectively cross-linked by phenolic acids. Obtained films showed higher mechanical parameters and thermal stability than films without phenolic acid addition. Moreover, such films showed novel bioactive properties such as antimicrobial and antioxidant activity.

The methods of obtaining materials, their physicochemical properties, and their antibacterial properties against different types of bacteria were reported and discussed. This review mainly provides information from recent years but also cites slightly older studies.

## Figures and Tables

**Figure 1 foods-12-01343-f001:**
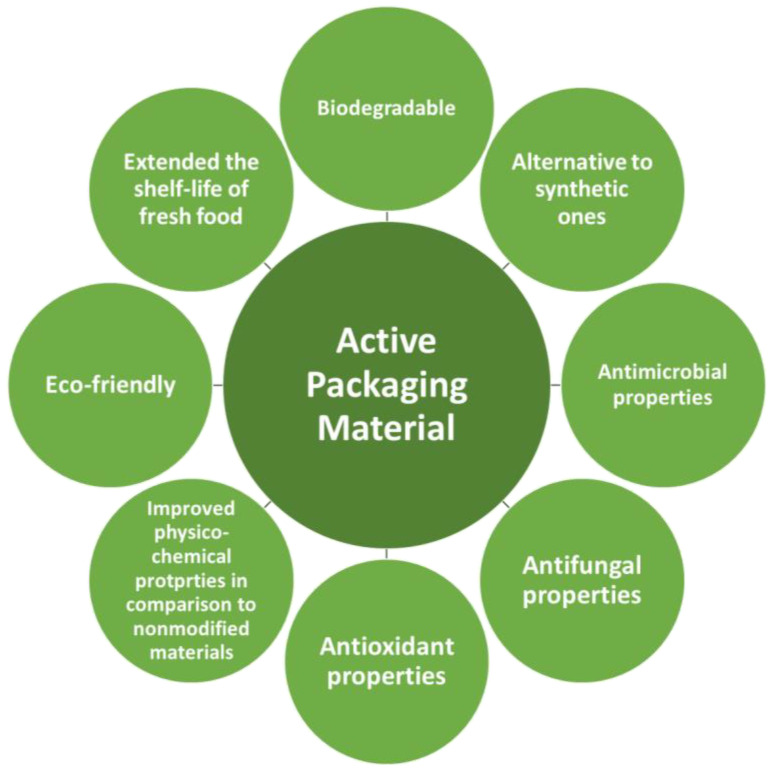
Characteristics of active packaging materials.

**Figure 2 foods-12-01343-f002:**
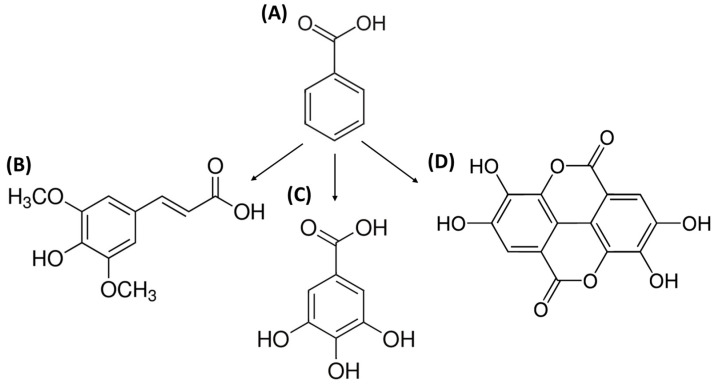
The structural formulas of benzoic acid (**A**) and its derivatives: sinapic acid (**B**), gallic acid (**C**), ellagic acid (**D**).

**Figure 3 foods-12-01343-f003:**
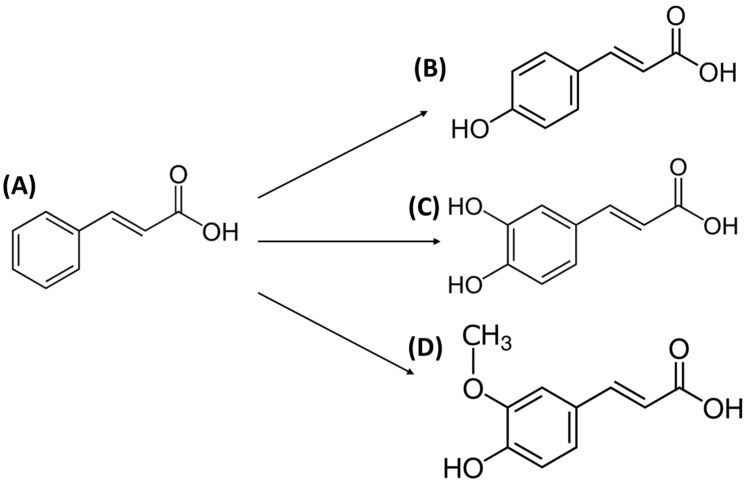
The structural formulas of cinnamic acid (**A**) and its derivatives: *p*-coumaric acid (**B**), caffeic acid (**C**), and ferulic acid (**D**).

**Table 1 foods-12-01343-t001:** Summary of biopolymeric materials modified with phenolic acids and their properties and applications.

Phenolic Acid	Polymer	Presented Properties after Phenolic Acid Addition	Application	Reference
Benzoic acid	Starch	Reduced the tensile strength by 24%; weakened optical properties; decrease in the whiteness index; increase in the yellowness index; reduction surface gloss; water vapor permeability without any changes	Edible films	[14]
	Chitosan	Ionic interactions between chitosan and benzoic acid; improved solubility of chitosan and benzoic acid; optically homogeneous transparent films	Packaging materials	[16]
	Starch/chitosan	Greater tensile strength; lower release rate of the preservative into model media; increased strength films by an average of 2 times; reduced release rate of the preservative; reduced vapor permeability of the films; antifungal activity against *A. niger*	Food-packaging materials	[15]
	Chitosan/whey protein	Stopped the development of pathogenic microorganisms: *Salmonella typhimurium*, *E. coli*, *C. jejuni*; protected turkey meat from microbiological deterioration	Edible films (to wrap fresh-cut turkey pieces)	[17]
	Protein from Argentine anchovy	Decreased tensile strength; increased elongation at break, increased color difference; increased opacity; increased water vapor permeability and solubility; homogeneous and continuous structure; presence of micropores on the film surface; antibacterial activity against *E. coli*, *S. enteritidis*, *L. monocytogenes*	Food-packaging films (bovine meat)	[18]
Sinapic acid	Chitosan	Antioxidant activity; antimicrobial activity against nine food-borne pathogens (*E. coli*, *S. aureus*, *P. microbilis*, *P. vulgaris*, *P. aeruginosa*, *E. aerogenes*, *B. thuringiensis*, *S. enterica*, *S. mutans*); no effect on thermal stability; higher soil degradability of chitosan-leaf films; lower soil degradability of chitosan-seed films	Edible films	[25]
	Lignin/chitosan	Antioxidant activity	Food-packaging films (protection of sensitive products from oxidation)	[24]
Gallic acid	Chitosan	Antioxidant activity against *E. coli*, *S. typhimurium*, *L. innocua*, *B. subtilis, S. aureus*; improved barrier properties; reduced water vapor and oxygen permeability; increased mechanical properties; homogeneous, transparent, and colorless films; higher thicknesses and water solubility	Food-packaging films (e.g., preservation of fresh pork); edible films; oxygen barrier films	[28,29,30,31,32,33]
	Chitosan with addition of ZnO	Enhanced mechanical properties; improved oxygen and water vapor permeability, swelling, water solubility, and UV–vis light transmittance; antioxidant activity; antibacterial activity against *B. subtilis* and *E. coli*	Food-packaging films	[34]
	Starch	Reduction of accessibility of starch molecules to digestive enzymes; antioxidant activity; antimicrobial activity; less tensile strength (0.9–7.8 MPa) with increasing plasticizer concentration (0.5–2.0 g glycerol/g starch); more homogeneous microstructures; increased water vapor permeability	Bioactive packaging films	[35,36,37,38]
	Chitosan/starch	Ester bonds, hydrogen bonds, and electrostatic interactions between starch, chitosan, and gallic acid; reduced elongation at brake; reduced film solubility in water; reduced water vapor permeability; improved tensile strength; excellent transparency of films	Edible films; packaging films	[39,40]
	Gelatin	Stronger tensile strength; higher elongation at brake; good antioxidant activity, biodegradability; antibacterial properties against *S. aureus* and *E. coli*; increased anti-ultraviolet capability	Packaging films	[41]
	Chitosan/gelatin	Reduction in transmittance; reduced elongation at the brake; reduced water vapor permeability; enhanced water vapor permeability; antioxidant activity; antimicrobial activity	Packaging films	[42]
	Gelatin/casein	Semi-crystalline structure; enhanced thermal stability; more homogenous surfaces; increased Young modulus and tensile strength; reduced elongation at the brake	Food-packaging materials	[43]
	Cellulose/ kappa carrageenan	Antioxidant activity; improved mechanical properties	Food-packaging films	[44]
Ellagic acid	Chitosan	Homogeneous, translucent, and flexible films; high mechanical parameters; high thermal stability; UVA and UVB barrier properties; water vapor permeability; high antioxidant activity; antibacterial activity against *S. aureus* and *P. aeruginosa*	Active eco-friendly packaging	[50]
	Starch	Rough surface with insoluble EA particles; modified tensile strength, elastic modulus, and elongation at break; films were capable of blocking UV light; high antioxidant activity	Packaging materials	[52]
	Sodium alginate	Intermolecular hydrogen bonding between the guava leaf extract and sodium alginate; enhanced antioxidant and antibacterial activity; enhanced tensile strength, water solubility, and water barrier properties; decreased moisture content and elongation at break	Green packaging films	[53]
Cinnamic acid	Starch	No significant influence on the barrier properties of the films; thermal stability; less water-soluble; more extensible; less resistant to break; antibacterial activity against *E. coli* and *L. innocua*	Food-packaging films (chicken breast and fresh-cut melon)	[63]
	Gelatin/chitosan	Antioxidant activity; bactericidal/bacteriostatic effect against *E. coli*; thermal stability	Bioactive packaging films	[68]
	Cellulose	Transparent films characterized by biodegradability, hydrophobicity, biosafety, and thermoplasticity; safe for human epidermal cells	Packaging materials (art paper, paper cups, paper straws, food-packaging boxes)	[71]
*p*-Coumaric acid	Chitosan	Antioxidant activity; UV light barrier ability; thermal stability; water vapor barrier ability; mechanical strength; antibacterial activity against *E. coli*, *S. typhimurium*, *S. aureus* and *L. monocytogenes*	Healthy food packaging	[77,79]
	Chitosan/gelatin	antioxidant activity; antimicrobial activity against *E. coli*, *Salmonella*, *B. subtilis* and *S. aureus*; thermal stability;	Packaging materials	[76]
	Cellulose nanocrystals/pectin	Coating barrier properties; antioxidant properties; decreased water vapor and oxygen permeability; inhibition effect of the fruit-browning process	Food-packaging films (fresh-cut fruits)	[80]
Caffeic acid	Chitosan	Antioxidant activity; lower solubility; wettability; thermal stability; less yellowness; better mechanical parameters	Packaging materials	[31,85]
	Gelatin	Decreased solubility; decreased water vapor permeability and oxygen permeability; barrier properties	Edible films	[86]
	Chitosan/ cellulose	Antioxidant activity; antimicrobial activity against *E. coli* and *S. aureus*; hydrophobicity, mechanical strength; water vapor barrier properties	Active food-packaging materials	[83]
	Chitosan/gelatin	Antioxidant activity; good mechanical properties; decreased the water vapor permeability; increased tensile strength	Edible films	[76,84]
Ferulic acid	Sodium alginate	Transparent and homogenous films; thermal stability; more rigid films than non-modified; antioxidant activity	Packaging films	[91]
	Chitosan	Antibacterial activity against *S. aureus* and *E. coli*; improved mechanical properties; improved thermal stability [92]; negative effect on mechanical properties [93]; positive effect on bioactivities	Packaging films	[92,93]
	Sodium alginate/ chitosan	High tensile strength; good light-blocking performance; hydrophobicity; thermal stability; lower water vapor transmission rate; lower swelling degree; strong interactions between the amino, carboxyl, and hydroxyl groups of the ferulic acid, sodium alginate, and chitosan	Packaging materials	[94]
	Pullulan/cellulose	Antioxidant activity; superior anti-fogging activity; high mechanical strength; thermal stability; water, oxygen, and carbon-dioxide barrier performances	Packaging films	[95]
	Collagen (with oxidized ferulic acid)	Good mechanical properties; thermal stability; resistance to enzyme degradation; anti-oxidant activity; antibacterial activity against *E. coli* and *S. aureus*	Active food-packaging materials	[104]
	Zein	Eliminated brittleness of films; extreme swelling; antioxidant activity; antimicrobial activity against *L. monocytogenes* and *C. jejuni*	Plasticizer	[101]
	Soy protein	Tensile strength and elongation at break; changes in film color and transparency; reduced water vapor permeability and water solubility; increased contact angle	Packaging materials	[102]
	Myofibrillar proteins of bigeye snapper	Enhanced mechanical properties; increased Young’s modulus and tensile strength; decreased elongation at break; decreased film transparency; barrier properties to UV light at the wavelength of 200–800 nm	Inner packaging material (for high-fat foods to prevent lipid oxidation)	[103]

**Table 2 foods-12-01343-t002:** Summary of other materials modified with phenolic acids and their properties and applications.

Phenolic Acid	Polymer	Presented Properties after Phenolic Acid Addition	Application	Reference
Benzoic acid	Poly(lactic acid)	Antibacterial activity against to *E. coli* and *S. aureus*; changes in film color	Antimicrobial food packaging	[19]
	Polyethylene	Antimycotic activity when in contact with media and cheese	Food packaging (cheese)	[20]
	Poly(ethylene-co-methacrylic acid)	Antimicrobial properties in fungal growth inhibition tests (*A. niger* and *Penicillium* sp.)	Antimicrobial food packaging films	[21]
Sinapic acid	Gelatin/Poly(vinyl alcohol)	Antioxidant activity; increased thickness of the films	Food packaging (cold-pressed vegetable oils)	[23]
Gallic acid	Poly(lactic acid)	Decreased surface hydrophilicity; decreased moisture sensitivity; improved barrier properties; good mechanical properties; reduced oxygen permeability	Food packaging (oils, biscuits, dry fruits, nuts, cereals)	[45]
	High-density polyethylene	Increased UV light stability; reduction of mechanical ductility and crystallinity; high contact transparency level	Food packaging	[46]
	Poly(ɛ-caprolactone)	Increased roughness of the surface; hydrophobicity; increased wettability; antioxidant activity; improved mechanical parameters	Antibacterial films	[47]
Cinnamic acid	Poly(lactic acid)	Worsened mechanical properties; less stiffness; less resistance to breaking; less extensible; improved water vapor and oxygen barrier capacity; thermal stability; no antibacterial activity against *E. coli* and *L. innocua*	Food packaging materials	[64,65,66]
	Poly(lactic acid)/starch	Effective growth inhibition of *E. coli* and *L. innocua*	Active packaging	[67]
	Poly(vinyl alcohol)	Better mechanical properties; better barrier properties; antioxidant activity; induced inhibition of *L. innocua* growth	Active packaging	[69,70]
	Konjac glucomannan/poly(lactic acid)	Antibacterial activity against *S. aureus* and *E. coli*; excellent mechanical properties; thermal stability; hydrophobicity; good swelling degree	Packaging materials	[72]
	Hemicellulose/ poly(vinyl alcohol)	Increased elongation at break; moderate oxygen barrier properties; good thermal stability; good UV barrier properties; antibacterial activity against *E. coli*	Packaging materials	[73]
*p*-Coumaric acid	Chitosan/poly(vinyl alcohol)/starch	Increased tensile strength; excellent swelling, water vapor transmittance, and antioxidant activity; thermal stability; antimicrobial activity against *E. coli* and *S. aureus*	Food-packaging films	[78]
	Chitosan/poly(vinyl alcohol)	Hydrophilic character of films; increased volume and weight swelling degree; decreased contact angle	Thin films	[81]
Caffeic acid	Chitosan/poly(lactic acid)	Lower permeability; higher fluidity; stronger ability to maintain free water;	New intelligent packaging	[88]
	Poly(vinyl alcohol- co-ethylene)	Increased mechanical parameters; positive radical scavenging activity; thermal stability	Food packaging	[87]
	Polycaprolactone	Hydrophobic behavior; decreased contact angle	Packaging films	[89]
Ferulic acid	Poly(lactic acid)	Increased thermal degradation temperature; limited release of FA from films; no significant antibacterial activity	Food packaging	[63,64,66]
	Ethylene vinyl alcohol copolymer	Enhanced ductility; thermal stability; UV-blocking effect; antimicrobial activity against *S. aureus* and *E. coli*; high effectiveness in the radical scavenging inhibition	Active food packaging	[96]
	Ethylene vinyl acetate copolymer	Antioxidant activity; thermal stability; better release behavior	Active packaging films with a controlled release of FA	[98]
	Poly(lactic acid)/poly(butylene adipate-co-terephthalate)	Antibacterial activity against L. monocytogenes and *E. coli;* slight tint of yellow; UV-light barrier property; enhanced tensile strength, fracture failure, elasticity; thermal stability	Active packaging films	[97]
	Poly(lactic acid)/starch	Antibacterial activity against *E. coli* and *L. innocua*; barrier properties	Food packaging	[67]
	Poly(lactic acid)/poly (vinyl alcohol)	Antibacterial activity against *E. coli* and *L. innocua*; thermal stability; good mechanical properties; barrier properties	Food packaging (meat preservation)	[99]
	Poly(lactic acid)/poly(3-hydroxybutyrate-co-3-hydroxyvalerate)	Higher glass transition temperature; thermal stability; stiffer and more resistant structure; improved oxygen and water vapor barrier capacity; antibacterial activity against *L. innocua* and *E. coli*	Food packaging	[100]

## Data Availability

The data presented in this study are available on request from the corresponding author.

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
