# Peer review of "The Application of Phenolic Acids in The Obtainment of Packaging Materials Based on Polymers—A Review"

_foods, 2023, doi:10.3390/foods12061343_

Round 1

Reviewer 1 Report

The review summarizes the materials based on polymers with phenolic acids, which is helpful in the application of phenolic acids in the obtainment of packaging materials based on polymers. Overall, the review shows comprehensive information about the topic. However, there are some issues should be addressed.

The following are the comments:

1. The authors summarized the characteristics of active packaging materials (shown in Fig. 1). In the case, they should show more details about those characteristics of the materials based on polymers with phenolic acids in the text and summarize them directly in Table 1.

2. It should avoid simply listing the results of others in the manuscript.

3. The comparison of the characteristics of the materials based on polymers with benzoic and cinnamic acid derivatives should be added in the manuscript, as they have distinct structural and functional differences.

There is still some room for improvement in writing. For example:

4. Line 30: ... a careful selections ?

5. Line 74: Benzoic acid has been defined as BA before (Line 71), so BA should be used in the following text. Please check throughout the manuscript.

6. Line 139: “It is very hard to find any articles about sinapic acid as a modifier to food packaging films.” can be deleted, as it has the same meaning as the previous one (it is not easy to find reports about its use as a modifier for biopolymers. Line 138).

7. Line 175: result in ...

Author Response

Reviewer 1:

On behalf of my co-authors and myself, I want to express our thanks for the valuable comments and constructive recommendations, which were very helpful for revising and improving our paper. We have studied the comments carefully before the revision of our article. All changes in the revised manuscript have been marked with the "Track Changes" function in Microsoft Word. We hope that the revised version of the manuscript will be acceptable for publication.

Comment 1: The authors summarized the characteristics of active packaging materials (shown in Fig. 1). In the case, they should show more details about those characteristics of the materials based on polymers with phenolic acids in the text and summarize them directly in Table 1.

Answer: Thank you very much for the comment. The information about the physico-chemical properties of discussed materials has been summarized in Table 1 (biopolymeric materials) and Table 2 (other materials). The characteristics of the materials based on the polymers modified with phenolic acids were supplemented and presented in tables.

Comment 2: It should avoid simply listing the results of others in the manuscript.

Answer: Thank you very much for the comment. In this review, we wanted to show the results of experimental studies related to polymeric-based materials with phenolic acids for the application as packaging materials. We wanted to show various compositions of films as well as the main results. We hope it is interesting for readers.

Comment 3: The comparison of the characteristics of the materials based on polymers with benzoic and cinnamic acid derivatives should be added in the manuscript, as they have distinct structural and functional differences.

Answer: Thank you very much for your comment. We delved deeper into the information about the properties describing materials based on polymers with different phenolic acids, and they are summarized in Tables 1 and 2. We hope that it will be the best way to present the properties of described materials.

Comment 4: Line 30: ... a careful selections ?

Answer: Thank you for the comment. The sentence is changed.

Comment 5: Line 74: Benzoic acid has been defined as BA before (Line 71), so BA should be used in the following text. Please check throughout the manuscript.

Answer: Thank you very much for the comment. We checked the manuscript carefully and the abbreviations of phenolic acids are used in following text.

Comment 6: Line 139: “It is very hard to find any articles about sinapic acid as a modifier to food packaging films.”can be deleted, as it has the same meaning as the previous one (it is not easy to find reports about its use as a modifier for biopolymers. Line 138).

Answer: Thank you very much for your comment. This repetition of the text has been removed.

Comment 7: Line 175: result in ...

Answer: Thank you. It is now corrected.

Reviewer 2 Report

This study summarizes and discusses studies about the potential of phenolic acids and using them in food packaging, their antioxidant and antimicrobial properties and interactions.  The paper does not really present novel concepts, ideas, tools, or data. We can find some similar papers in literature such as “Biodegradable active materials containing phenolic acids for food packaging applications, COMPREHENSIVE REVIEW, 2022” or “Phenolic Compounds in Active Packaging and Edible Films/Coatings: Natural Bioactive Molecules and Novel Packaging Ingredients, molecules, 2022”.

The conclusion is summary of the manuscript and repeats the texts . So this paper didn’t present new findings in the area.  

Author Response

Reviewer 2:

On behalf of my co-authors and myself, I want to express our thanks for the valuable comments and constructive recommendations, which were very helpful for revising and improving our paper. We have studied the comments carefully before the revision of our article. All changes in the revised manuscript have been marked with the "Track Changes" function in Microsoft Word. We hope that the revised version of the manuscript will be acceptable for publication.

Comment 1: This study summarizes and discusses studies about the potential of phenolic acids and using them in food packaging, their antioxidant and antimicrobial properties and interactions.  The paper does not really present novel concepts, ideas, tools, or data. We can find some similar papers in literature such as “Biodegradable active materials containing phenolic acids for food packaging applications, COMPREHENSIVE REVIEW, 2022” or “Phenolic Compounds in Active Packaging and Edible Films/Coatings: Natural Bioactive Molecules and Novel Packaging Ingredients, molecules, 2022”.

The conclusion is summary of the manuscript and repeats the texts . So this paper didn’t present new findings in the area. 

Answer: Thank you very much for the comment. The aim of our review was to summarize the knowledge about polymeric-based materials with phenolic acids addition for the application as food packaging.

The authors of the paper “Biodegradable active materials containing phenolic acids for food packaging applications, COMPREHENSIVE REVIEW, 2022” focused only on biodegradable polymers. They did not include all polymers that we discussed. Also, the composition of the review is different as their discussion is the concentration of polymers modification, whereas in our paper, we discuss different phenolic acids with examples of polymers. We understand that review is similar, but its composition, as well as content, is not the same.

The similarities of the paper “Phenolic Compounds in Active Packaging and Edible Films/Coatings: Natural Bioactive Molecules and Novel Packaging Ingredients, molecules, 2022” to our can be found. However, our paper has a completely different composition. We discussed different phenolic acids and described their addition to various polymers. In the cited paper, authors discussed selected properties of materials and treated phenolic acids mostly as groups.

In our opinion, we show a different views on the modifications of polymeric-based materials by phenolic acid addition. The presence of similar reviews from 2022 shows how important and the novel is the content of our review.

Reviewer 3 Report

To meet the publication requirements of the journal, the authors should address the following five questions.
1. Table 1 is not appropriate in the conclusion section and should be placed after Section 4.
2. Table 1 should summarize the role of phenolic acids in packaging, such as antibacterial, antioxidant, and enhancing the physical and chemical properties of polymers, and introduce the application of packaging containing phenolic acids, such as fresh-keeping of fruits and vegetables.
3. The author should properly describe the future application potential of phenolic acid/polymers packaging and the possibility of industrialization in the future perspectives section.
4. The conclusion should be rewritten because it does not summarize the main point of the article.
5. “Antioxidant properties” should be added as keywords.

Author Response

Reviewer 3:

On behalf of my co-authors and myself, I want to express our thanks for the valuable comments and constructive recommendations, which were very helpful for revising and improving our paper. We have studied the comments carefully before the revision of our article. All changes in the revised manuscript have been marked with the "Track Changes" function in Microsoft Word. We hope that the revised version of the manuscript will be acceptable for publication.

Comment 1: Table 1 is not appropriate in the conclusion section and should be placed after Section 4.

Answer: Thank you very much for the comment. We prepared completely new section to summarize described materials, together with properties and applications of them. Additionally we shared the table for two separate parts. First table is summarizing biopolymeric materials and second table is summarizing information about other described materials.

Comment 2: Table 1 should summarize the role of phenolic acids in packaging, such as antibacterial, antioxidant, and enhancing the physical and chemical properties of polymers, and introduce the application of packaging containing phenolic acids, such as fresh-keeping of fruits and vegetables.

Answer: Thank you very much for your comment. Good point. Right now, the summary of the materials and their properties and potential applications were summarized in table 1 and table 2 in section 5.

Comment 3: The author should properly describe the future application potential of phenolic acid/polymers packaging and the possibility of industrialization in the future perspectives section.

Answer: Thank you for the comment. We added information about the need to transfer the scale.

Comment 4: The conclusion should be rewritten because it does not summarize the main point of the article.

Answer: Thank you very much for the comment. “Conclusions” section is now improved.

Comment 5: “Antioxidant properties” should be added as keywords.

Answer: Thank you very much for your comment. This phrase was added to keywords.

Round 2

Reviewer 1 Report

The manuscript is improved.

Reviewer 2 Report

As it was said before , the conclusion is summary of the manuscript and repeats the texts  and this paper didn’t present new findings in the area.  

I recommend this paper to reject again.